# Metabolomic Profiling in Combination with Data Association Analysis Provide Insights about Potential Metabolic Regulation Networks among Non-Volatile and Volatile Metabolites in *Camellia sinensis cv Baijiguan*

**DOI:** 10.3390/plants11192557

**Published:** 2022-09-28

**Authors:** Mingjie Chen, Xiangrui Kong, Yi Zhang, Shiya Wang, Huiwen Zhou, Dongsheng Fang, Wenjie Yue, Changsong Chen

**Affiliations:** 1College of Life Sciences, Henan Provincial Key Laboratory of Tea Plant Biology, Xinyang Normal University, Xinyang 464000, China; 2Tea Research Institute, Fujian Academy of Agricultural Sciences, Fuzhou 350012, China; 3School of Life Sciences, Nanchang University, Nanchang 330031, China; 4Jinshan College, Fujian Agriculture and Forestry University, Fuzhou 350002, China

**Keywords:** *Camellia sinensis*, albino tea, half-sibs, catechins, amino acids, MVA pathway, MEP pathway, phenylpropanoid pathway, fatty acid-derivative pathway, correlation analysis

## Abstract

The non-volatile and volatile metabolites in tea confer the taste and odor characteristics of tea fusion, as well as shape the chemical base for tea quality. To date, it remains largely elusive whether there are metabolic crosstalks among non-volatile metabolites and volatile metabolites in the tea tree. Here, we generated an F1 half-sib population by using an albino cultivar of *Camellia sinensis cv Baijiguan* as the maternal parent, and then we quantified the non-volatile metabolites and volatile metabolites from individual half-sibs. We found that the EGC and EGCG contents of the albino half-sibs were significantly lower than those of the green half-sibs, while no significant differences were observed in total amino acids, caffeine, and other catechin types between these two groups. The phenylpropanoid pathway and the MEP pathway are the dominant routes for volatile synthesis in fresh tea leaves, followed by the MVA pathway and the fatty acid-derivative pathway. The total volatile contents derived from individual pathways showed large variations among half-sibs, there were no significant differences between the albino half-sibs and the green half-sibs. We performed a comprehensive correlation analysis, including correlations among non-volatile metabolites, between volatile synthesis pathways and non-volatile metabolites, and among the volatiles derived from same synthesis pathway, and we identified several significant positive or negative correlations. Our data suggest that the synthesis of non-volatile and volatile metabolites is potentially connected through shared intermediates; feedback inhibition, activation, or competition for common intermediates among branched pathways may co-exist; and cross-pathway activation or inhibition, as well as metabolome channeling, were also implicated. These multiple metabolic regulation modes could provide metabolic plasticity to direct carbon flux and lead to diverse metabolome among *Baijiguan* half-sibs. This study provides an essential knowledge base for rational tea germplasm improvements.

## 1. Introduction

Tea is the second popular beverage worldwide after water, largely due to its enjoyable taste, elegant flavor, and health benefits. These attributes are conferred by its rich non-volatile and volatile metabolites. The major non-volatile metabolites in tea include polyphenols (mainly catechins), amino acids (mainly theanine), caffeine, vitamins, and carbohydrates, which collectively contribute to the taste of tea infusion [1,2]. Amino acids are the major sources of umami taste, while polyphenols and caffeine confer the astringency and bitterness of tea infusion [3]. Thus, higher levels of free amino acids endow tea with an umami taste; lower levels of caffeine and catechins reduce tea’s astringency and bitterness [3,4]. The odor quality of tea is determined by the amounts and compositions of various volatiles present in it. To date, more than 700 tea volatiles have been reported [5,6,7,8]. Due to their large variations in odor threshold and contents in tea, not all volatiles contribute to the fragrance of tea infusion [9]. The volatiles present in the made tea are either retained from fresh tea leaves or newly generated during tea processing. The endogenous volatiles retained from fresh tea leaves make significant contributions to cultivar-specific aroma characteristics of made tea [9]. Thus, tea germplasm genetic improvement is essential for the quality improvement of tea products.

Recently, tea leaf color variations (white, yellow, or purple) have attracted widespread attention for tea tree germplasm improvement. Based on their response to environmental factors, albino tea germplasms are divided into three types: temperature-sensitive, light-sensitive, and temperature/light-dual-sensitive [10]. The temperature-sensitive albino teas exhibit white albino shoots during early spring when the atmosphere temperature is below 20 °C, and they turn green when temperatures rise above 22 °C [11]. “Xiaoxueya”, “Anjibaicha”, and “Huabai 1” are of this type [12,13,14]. The light-sensitive albino teas show yellow albino shoots under open field conditions, and they turn green under shading conditions. Such germplasms include “Jinguang”, “Yujinxiang”, “Huangjinya”, and “Yinghong No.9 yellow mutant” [15,16,17,18,19]. “Baijiguan” is a unique albino landrace originating from the Wuyi mountain of Southeast China, and it exhibits light/temperature-dual-sensitivity [10,16]. Leaf color variations significantly affect the accumulations of catechin and anthocyanin [14,20,21,22,23,24], theanine [11,25,26,27,28], caffeine [3], and sugars [29]. In addition, leaf color variations also affect tea volatile metabolites [3,22,30]. These reports raise an intriguing question regarding whether non-volatile and volatile tea metabolic pathways interact in vivo. Understanding the underlying mechanisms is fundamental for rational tea germplasm improvements since both non-volatile and volatile traits are the main targets for tea tree breeding programs. 

Gene co-expression analysis strongly suggests that volatile and non-volatile metabolisms are intimately interconnected. In *Arabidopsis*, the genes for indole, phenylpropanoid, and flavonoid biosynthesis were found to be co-expressed [31], and genes for isoprenoid biosynthesis pathways were also found to be co-expressed [32]. In tomato fruit, the overexpression of SIMYB75 increases the contents of anthocyanin and phenylpropanoid-derived and terpene volatiles [33]. Even though the tea tree is a cash crop, few studies have provided an integrated view of its metabolic interactions, partially due to its self-incompatibility, which leads to a high genetic heterogeneity. To overcome these barriers, two different strategies have been widely applied. The first one relies on another green tea cultivar as control [17,25,29,34,35,36]. The drawback of this method is that the selected control green tea cultivar is genetically unrelated to the albino tea under study. The second strategy is using the same albino tea germplasm as the control, but it is grown under different environmental conditions (shading, lower temperature, etc.) such that the albino leaf shows visible color changes [12,14,16,17,18,19,24,26,29,37,38,39,40,41,42]. The limitation of this method is that the data are complicated by environmental effects that usually exceed the genetic ones. Although the use of these methods has offered multiple important insights regarding the molecular mechanisms underlying tea leaf albinism, the metabolic connectivity between non-volatile and volatile metabolism remains largely unresolved due to these limitations. 

*Camellia sinensis cv Baijiguan* is a heritable albino tea cultivar; its half-sibs exhibit green or yellow leaf color under identical growth conditions (Appendix A). Due to their similarity in genetic makeup, this half-sib population provide an ideal system to dissect the relationships between volatile and non-volatile metabolism. In this study, non-volatile and volatile metabolites were quantitatively measured from *Baijiguan* half-sibs, and association analysis was then conducted to explore the correlations between non-volatile metabolites and volatile synthesis pathways, as well as the correlations among metabolites derived from the same synthesis pathway. This research offers novel insights regarding potential metabolic regulation networks in tea plants, and it could facilitate the rational design of new tea germplasms with desirable metabolic traits for tea quality improvements.

## 2. Results

### 2.1. Non-Volatile Contents of the Albino Half-Sibs and the Green Half-Sibs of Baijiguan

Non-volatile tea metabolites, including total amino acids, caffeine, gallic acid (GA), (+)-catechin (C), (−)-epicatechin (EC), (+)-gallocatechin (GC), (−)-epigallocatechin (EGC), (−)-epicatechin gallate (ECG), catechin gallate (CG), gallocatechin gallate (GCG), and (−)-epigallocatechin gallate (EGCG), were quantitatively measured from the albino half-sibs and the green half-sibs of *Baijiguan*; the full dataset is provided in Appendix A. The EGC, EGCG, total non-ester catechins, and total ester catechins contents from the albino half-sibs were significantly lower than those of the green half-sibs (Figure 1). The total amino acids, caffeine, and other catechin types did not show statistically significant differences between the albino half-sibs and the green half-sibs (Appendix A). 

### 2.2. Volatile Contents of the Albino Half-Sibs and the Green Half-Sibs of Baijiguan

The volatiles were isolated and quantitatively measured from the albino half-sibs and the green half-sibs of *Baijiguan*. Based on their origins, these volatiles can be divided into four groups: volatiles derived from the phenylpropanoid pathway, the methylerythritol phosphate (MEP) pathway, the mevalonate (MVA) pathway, and the fatty acid-derivative pathway. The volatiles derived from the same pathway were subtotaled, and the data are presented in Table 1. The total phenylpropanoid volatile contents ranged from 4.35 µg. g^−1^ FW (0317N) to 37.24 µg. g^−1^ FW (0309A); the total monoterpene/diterpene contents ranged from 3.06 µg. g^−1^ FW (0317N) to 31.00 µg. g^−1^ FW (0317D). Regardless leaf color, these two pathways collectively accounted for ~90% of the total fresh leaf volatile contents. The sesquiterpene volatiles synthesized through the MVA pathway ranged from 0.75 µg. g^−1^ FW (0317N) to 3.19 µg. g^−1^ FW (0306B), and they accounted for 2% to 10% of total leaf volatile contents. The fatty acid-derivative volatiles were the least numerous and accounted for only 0.1% to 2.3% of total tea leaf volatile contents (Table 1). Even though there were large variations among individual half-sibs, there was no significant difference between the albino half-sibs and the green half-sibs for the total volatile contents derived from any individual pathway (Appendix A). 

### 2.3. Correlations among Non-Volatile Metabolites of Baijiguan Half-Sibs

Correlations among non-volatile metabolites including total amino acids, caffeine, and various catechins were analyzed, and several significant correlations were found among catechins (Figure 2a). Thus, the catechin synthesis pathway was analyzed in detail. The catechin synthesis pathway is first branched at dihydrokaempferol (indicated by A in Figure 2b), and we named these two branches the R1 and R2 routes (Figure 2b). The R1 route is further branched at leucocyanidin (indicated with B in Figure 2b), and we named them R1-1 and R1-2 routes. The R1-1 route leads to the synthesis of EC and ECG, and the R1-2 route leads to the synthesis of C and CG. The R2 route is branched at leucodelphinidin (indicated by C in Figure 2b), and we named them R2-1 and R2-2 routes. The R2-1 route leads to the synthesis of GC and GCG, and the R2-2 route is responsible for EGC and EGCG synthesis (Figure 2b). Based on their stable levels (Appendix A), the carbon flux was calculated and is presented by arrow size (Figure 2b). The carbon flux through the R1 route was about 2–4 fold lower than that of the R2 route. Within the R1 route, the carbon flux through the R1-1 route was 4-fold higher than that of the R1-2 route. Within the R1-1 route, more than 67% of EC was esterified into ECG, while a variable amount of C was converted into CG through the R1-2 route. Within the R2 route, the carbon flux through the R2-2 route was 4-fold higher than that of the R2-1 route. Within the R2-2 route, more than 80% of EGC was esterified into EGCG (Figure 2b). GA is synthesized from the shikimate pathway intermediate 3-dehydroshikimate [43]. Correlation analysis showed that GA was positively correlated with EGC (R = 0.64), EGCG (R = 0.56), non-ester catechins (R = 0.69), and ester catechins (R = 0.61) (Figure 2a). Since GA and EGC synthesis occur upstream and the downstream of highly branched pathways, respectively (Appendix A), their positive correlation suggest some synergism to coordinate the synthesis of GA and EGC. The availability of 3-dehydroshikimate is regulated by bifunctional enzyme 3-dehydroquinate dehydratase/shikimate dehydrogenase (DHQD/SD) [44]. DHQD catalyzes 3-dehydroquinate into 3-dehydroshikimate, while SD converts 3-dehydroshikimate into D-shikimate. In plants, the enzyme activity of DHQD is 10 times higher than SD activity [45]. Thus, 3-dehydroshikimate is accumulated, and then dehydroshikimate dehydrogenase (DSDG) converts it into GA [46]. Unexpectedly, the stable GA levels were found to be rather low in *Baijiguan* half-sibs (Appendix A). Although GA could be methylated into methyl gallate (MG) [47], the MG level was even lower than that of GA. Thus, GA methylation would not be the major factor to keep free GA level low in planta. Our data suggest that the DSDG enzyme cannot efficiently convert 5-dehydroshikimate into GA (Appendix A); this low efficiency may due to its reportedly low affinity for 5-dehydroshikimate (K_m_ = 0.49 mM) and high affinity for NADP^+^ (K_m_ = 0.008 mM) [46]. Thus, only when the carbon flux into the shikimate pathway is abundant could GA synthesis be sustained. In addition, GA synthesis could be regulated by the redox status of the plant cell. Since GA was positively correlated with EGC synthesis (Figure 2a), we speculate that GA availability could be the major bottleneck to control EGCG synthesis in response to carbon flux. The positive correlation between GA and EGC may have dual roles in metabolic regulation: (1) keep the free GA level low as it may be toxic at higher concentrations and (2) provide sufficient free GA for EGCG synthesis when the carbon flux into the shikimate pathway is abundant. 

Both positive correlations and negative correlations were identified between the R1 and R2 routes. GCG was negatively correlated with EC (R = −0.64) and ECG (R = −0.70) (Figure 2a,b). Considering that GCG is the product of a minor carbon flux route (R2-1), its level would be more sensitive to the R2 route carbon flux changes compared with the R2-2 route product EGCG. EC and ECG are synthesized through the R1-1 route, which is the dominant branch route within the R1 route. On the one hand, the negative correlations of GCG to both EC and ECG likely reflect the competition of R1 and R2 routes for the restricted common substrate dihydrokaempferol; on the other hand, these negative correlations could also provide a sensitive mechanism for cross-pathway feedback inhibition. 

Between the R1 and R2 routes, GC was negatively correlated with C (R = −0.64) but positively correlated with CG (R = 0.55) (Figure 2a,b). Interestingly, the R1-2 route (for C and CG synthesis) and the R2-1 route (for GC synthesis) were found to be the minor carbon flux routes of the R1 and R2 routes, respectively; we reasoned that their level could be more sensitive to the carbon flux changes of the R1 and R2 routes. The presence of both positive and negative correlations may reflect the co-existence of cross-pathway activation and inhibition, which could fine-tune the carbon flux between the R1 and R2 routes. 

Within the R1 route, we found no apparent correlation between the R1-1 and R1-2 routes. However, within the R1-1 route, ECG was positively correlated with its precursor EC (R = 0.70), suggesting that EC supply within this route is sufficient in planta. In contrast, within the R1-2 route, CG was negatively correlated with its precursor C (R = −0.6), suggesting that C supply within this route is restricted in planta. 

Within the R2 route, EGC from R2-2 route was negatively correlated with GCG from the R2-1 route (R = −0.61) (Figure 2a,b), suggesting that EGC suppresses carbon flux into R2-1 such that it maximizes its own level. Interestingly, EGC was positively correlated with EGCG in the R2-2 route (R = 0.8) (Figure 2a,b), which suggests that EGCG is the predominant catechin in the R2 route and in planta.

### 2.4. Correlations among the Four Volatile Synthesis Pathways in Baijiguan Half-Sibs

Among the four volatile synthesis pathways, the MVA pathway was positively correlated with the fatty acid-derivative pathway (R = 0.68) (Figure 2a). Accordingly, HCA and heap map analysis grouped the MVA volatile synthesis pathway and the fatty acid-derivative pathway in the same cluster (Figure 3a). Previous studies demonstrated that fatty acid catabolism and synthesis take place concurrently in planta [48,49]. Although the physiological function of fatty acid catabolism under normal growth conditions remains unclear, its degradation products (*cis*-3-hexenol, hexanoate, and hexadecenoic acid) can serve as substrates for the synthesis of FA-derivative volatiles (*cis*-3-hexenyl hexanoate, *cis*-jasmone, and hexadecenoic acid methyl ester), as well as the MVA pathway precursor acetyl-CoA. At the metabolic level, the MVA pathway has three compartments (cytosol, ER, and peroxisome). In peroxisome, the β-oxidation of fatty acids produces acetyl-CoA, a peroxisome-localized acetoacetyl-CoA thiolase that condenses acetyl-CoA into acetoacetyl-CoA [50,51,52,53]. The positive correlations between the MVA pathway and the FA-derivative pathway raise an open question regarding whether the acetoacetyl-CoA subpool produced through FA β-oxidation in peroxisome could be channeled into the MVA pathway for sesquiterpene volatile synthesis in tea leaves.

### 2.5. Correlations between Volatile Synthesis Pathways and Non-Volatile Metabolites in Baijiguan Half-Sibs

The correlations analysis of the four volatile synthesis pathways and the major non-volatile tea metabolites (total amino acids, caffeine, and catechins) identified several positive correlations and negative correlations. For example, the phenylpropanoid volatile pathway was positively correlated with EGCG (R = 0.65) (Figure 2a). The synthesis of phenylpropanoid volatiles and EGCG was shown to be derived from shikimate or phenylalanine (Appendix A). This positive correlation suggests that the carbon flux through the shikimate pathway is abundant in tender tea leaves. To support this notion, it is estimated that in land plants, from 20% to 50% of fixed carbon passes through the shikimate pathway [44,54,55]. A second positive correlation was observed between the MVA volatile pathway and ECG (R = 0.58) (Figure 2a). Both MVA volatile synthesis and ECG synthesis share a common precursor, cytosolic acetyl-CoA [51]. Acetyl-CoA acetyltransferase or acetoacetyl-CoA thiolase (AACT) convert cytosolic acetyl-CoA into acetoacetyl-CoA, which commit to sesquiterpene volatile synthesis; meanwhile, cytosolic ACCase converts acetyl-CoA into malonyl-CoA for the synthesis of chalcone, which is a common upstream intermediate for ECG and other catechin syntheses (Figure 3b and Appendix A). When the cytosolic acetyl-CoA supply is sufficient, MVA volatiles are expected to be positively correlated with ECG (Figure 2a). Like ECG, other catechins (GCG and EGCG) are also derived from cytosolic acetyl-CoA (Figure 2b). However, GCG or EGCG did not show significant correlations with the MVA volatile pathway (Figure 2a). This discrepancy raises an open question regarding whether a specific cytosolic acetyl CoA subpool is channeled to MVA volatile synthesis and non-pyrogallol catechin (ca. ECG) synthesis in the tea tree.

In addition to the positive correlations discussed above, a negative correlation was identified between the MVA volatile pathway and total amino acid contents (R = −0.65) (Figure 2a). Accordingly, both are grouped into different clusters in Figure 3a. To explain this negative relationship, we propose a working model in Figure 3b. In this model, cytosolic acetyl-CoA could be generated from amino acid catabolism through two routes: (1) Aspartate-derived amino acids (lysine, methionine, threonine, and isoleucine) are known to be degraded in mitochondria to produce acetyl-CoA, and this acetyl-CoA is converted into citrate [56,57]. The citrate could be exported into cytoplasm, where a cytosolic ATP-citrate lyase transforms citrate into acetyl-CoA. This carbon diversion would reduce the α-ketoglutarate supply [58], thus reducing glutamate synthesis via Gln conversion (Figure 3b). (2) Theanine degradation also contributes to cytosolic acetyl-CoA synthesis through released ethylamine oxidation. Free amino acids in tea are mainly constituted of glutamate, glutamine, aspartic acid, and L-theanine [59]. Glutamate is also the precursor for glutamine and L-theanine synthesis [60,61]. Thus, the catabolism of theanine and aspartate-derived amino acids is expected to reduce the total amino acid contents but increase the cytosolic acetyl-CoA supply for enhanced MVA volatile synthesis (Figure 3b). Recently, Huang et al. (2022) conducted a quantitative trait loci (QTL) mapping by using an F1 full-sib population derived from *Baijiguan* and *Longjing 43* [62], and they identified multiple QTLs for free amino acid contents. Thus, free amino acid contents are polygenic traits in the tea tree. Since the albino leaf phenotype of *Baijiguan* was a single-gene dominant trait [10], it is expected that the QTLs for total free amino acid content were not fully overlapped with the albino phenotype of *Baijiguan*. This may explain why there was no significant difference in total amino acid contents between the albino half-sibs and the green half-sibs (Appendix A). 

### 2.6. Correlations among the Volatiles Derived from MVA Pathway in Baijiguan Half-Sibs

Considering that enzyme reactions observe stoichiometry within the same synthesis pathway, the metabolite composition within the same pathway implies underlying enzyme reactions or regulations. Thus, we converted volatile content data in Appendix A (μg. g^−1^ FW) into Mol% (Appendix A), and then we used the Mol% data for correlation analysis. Although the carbon flux through the four volatile synthesis pathways all exhibited large variations among half-sibs (Table 1), the Mol% data transformation essentially suppresses the differences in total carbon flux among half-sibs. 

The HCA and heat map analysis of all the 14 sesquiterpenes (Appendix A) are presented in Figure 4a. Correlation analysis uncovered significant positive and negative correlations among individual member of the MVA pathway (Appendix A). Based on the connectivity of the correlations, these volatiles were grouped into three clusters. Cluster I included α-farnesene, β-farnesene, α-muurolene, and germacrene B. Except for a negative correlation between β-farnesene and germacrene B (R = −0.62), α-farnesene, β-farnesene, and α-muurolene were mutually positively correlated (Figure 4b). CsAFS catalyzes α-farnesene synthesis [63], though the genes responsible for β-farnesene, α-muurolene, and germacrene B synthesis have not been identified in the tea tree. The mutual positive correlations among α-farnesene, β-farnesene, and α-muurolene suggest some common mechanisms for their synthesis. 

Cluster II included α-copaene, δ-cadinene, and β-caryophyllene; δ-cadinene was positively correlated with α-copaene (R = 0.80) and β-caryophyllene (R = 0.59) (Figure 4b). Some TPSs are reported to be multiple-product synthases. For example, the recombinant *Arabidopsis* TPS (At5g23960) protein catalyzes FPP into α-copaene, β-caryophyllene, and other sesquiterpenes [64]; VvShirazTPS26 converts FPP into α-copaene, δ-cadinene, and other sesquiterpenes [65]. The TPSs responsible for α-copaene, δ-cadinene, and β-caryophyllene synthesis in the tea tree have not been reported. The positive correlations among these three sesquiterpenes raise an open question regarding whether a tea TPS homolog of At5g23960 or VvShirazTPS26 could be responsible for these volatile syntheses. 

Cluster III included β-caryophyllene oxide, α-humulene, α-cadinol, cubebol, α-cadinene, and germacrene D. Except for a positive correlation with germacrene D (R = 0.68), α-cadinol showed negative correlations with α-cadinene (R = −0.8), cubebol (R = −0.66), and α-humulene (R = −0.64). α-humulene was negatively correlated with β-caryophyllene oxide (R = −0.62) (Figure 4b). Cop2 is a multi-product terpene synthase for α-cadinol and germacrene D synthesis [66]. The positive correlation between α-cadinol and germacrene D opens the possibility that a functional tea tree homolog of Cop2 exists and is responsible for their synthesis. Both α-cadinol and cubebol were proposed to be derived from the cadinyl cation; similarly, both β-caryophyllene oxide and α-humulene were proposed to be derived from the humulyl cation (Figure 4c). The negative correlations between these two pairs of volatiles suggest that the formation of cadinyl and humulyl cations is limited in tea leaves (Figure 4b,c).

### 2.7. Correlations among the Volatiles Derived from MEP Pathway in Baijiguan Half-Sibs

The results of the HCA and heat map analysis of the 12 monoterpenes and diterpenes are presented in Figure 5a. Correlation analysis identified both positive correlations and negative correlations among them (Appendix A). The types of correlations and the connectivities are summarized in Figure 5b. Overall, the MEP pathway volatiles can be divided into two clusters through linalool synthase (Figure 5b,c). Cluster I included upstream monoterpenes and diterpenes of the pathway, and cluster II included linalool and its oxides, which are located the downstream of the MEP pathway (Figure 5c). Since (R)-linalool and (S)-linalool could not be separated under our assay conditions, the linalool contents represent the sum of (S)-linalool and (R)-linalool in Appendix A. Based on the stable levels of individual volatile contents (Appendix A), about 60–90% of carbon was channeled into linalool synthesis, suggesting that CsSRIS, CsRLIS, or CsLIS/NES-1 can efficiently pull carbon into linalool synthesis. Linalool enantiomers are oxidized by P450s into hydroxylated or epoxidized products [67,68]. The furanoid isomers of *trans*-linalool oxide and *cis*-linalool oxide are converted into their respective pyranoid isomers through unknown mechanism(s) (Figure 5c). Since the furanoid isomer contents of linalool oxide were higher than those of their respective pyranoid isomers (Appendix A); this transformation reaction is not efficient in the tea tree.

Correlation analysis showed that the cluster I metabolites were positively correlated. For example, cyclic monoterpene α-terpinene was positively correlated with acyclic monoterpene neral (R = 0.76), and both were positively correlated with diterpene neophytadiene and geranyllinalool (Figure 5b). Two acyclic monoterpenes—(E)-β-ocimene and nerol—were also positively correlated (R = 0.84) (Figure 5b). Within cluster II, only linalool and *trans*-linalool oxide (furanoid) were positively correlated (R = 0.94) (Figure 5b). Between cluster I and cluster II, opposite types of correlations were observed: linalool, *trans*-linalool oxide (furanoid), and *trans*-linalool oxide (pyranoid) showed negative correlations with multiple cluster I metabolites; in contrast, *cis*-linalool oxide (furanoid) and *cis*-linalool oxide (pyranoid) were positively correlated with geranyllinalool and linalool structural isomer geraniol (Figure 5b). Linalool oxides have been proposed to play diverse roles, including in the synthesis of allelochemicals for plant–microbe or plant–insect interactions, linalool detoxification, monoterpenol solubility enhancement through oxidation and subsequent glycosylation, enhanced sequestration, and transport to other organs [69,70,71,72]. Our data suggest that *trans*- and *cis*-linalool oxide stereoisomers could function as feedback inhibitors and activators, respectively (Figure 5c). The opposite effects of *trans*- and *cis*-linalool oxide stereoisomers may provide versatility to fine-tune carbon flux into cluster I and cluster II products.

### 2.8. Correlations among the Volatiles Derived from the Shikimate–Phenylpropanoid Pathway and Fatty Acid-Derivative Pathway

For shikimate–phenylpropanoid volatiles, indole was positively correlated with methyl salicylate (R = 0.8) (Figure 6a). Indole and methyl salicylate are synthesized in chloroplast, and they use common intermediate chorismic acid through branched pathways (Appendix A) [73,74]. As we discussed above, the shikimate pathway is a high flux-bearing pathway [44], and it is expected that there is a sufficient chorismic acid supply to fuel these branched pathways, thus leading to the positive correlation between indole and methyl salicylate (Figure 6a). Although phenylethyl alcohol and benzyl alcohol are synthesized from their common intermediate phenylalanine through branched pathways, both were negatively correlated (R = −0.74) (Figure 6a). For phenylethyl alcohol synthesis, phenylalanine is first converted into phenylacetaldehyde by phenylacetaldehyde synthase (PAAS), and then phenylacetaldehyde reductase (PAR) reduces phenylacetaldehyde into phenylethyl alcohol; for benzyl alcohol synthesis, phenylalanine is first converted into *trans*-cinnamic acid (*trans*-CA) by phenylalanine ammonia lyase (PALs) [75], and then *trans*-CA is converted into benzaldehyde and benzyl alcohol (Appendix A). Besides serving as a precursor for benzyl alcohol synthesis, *trans*-CA is also the sole precursor for the biosynthesis of lignin, flavonoids, and phytoalexin, which are the major carbon flux routes in planta [76]. PALs are highly efficient enzymes [77], and most phenylalanine is expected to be channeled into *trans*-CA synthesis. In the tea tree, when *PAL* transcription is induced by blue or red light, volatile benzenoids are significantly increased [78], suggesting that *PALs* are the major genes regulating benzyl alcohol synthesis. Interestingly, *PALs* are subjected to multiple transcriptional regulations [79,80,81,82]. Since PALs simultaneously decrease phenylalanine levels and increase *trans*-CA contents, this could affect phenylethyl alcohol and benzyl alcohol synthesis in opposite directions. A negative correlation between benzyl alcohol and phenylethyl alcohol, as demonstrated here, further supports PALs’ role in regulating phenylethyl alcohol and benzyl alcohol synthesis (Figure 6a).

Among the fatty acid-derivative volatiles, hexadecenoic acid methyl ester was negatively correlated with *cis*-3-hexenyl hexanoate (R = −0.91) and *cis*-jasmone (R = −0.99). In contrast, *cis*-3-hexenyl hexanoate was positively correlated with *cis*-jasmone (R = 0.85) (Figure 6b). Considering that hexadecenoic acid (16:1) is present in fresh tea leaves at an appreciable amount (~59 μg g^−1^ fresh weight) [83], these negative correlations raise an open question regarding whether hexadecenoic acid could be oxidized into hexanol, hexanate, or *cis*-jasmone in tea plants. 

## 3. Discussion

### 3.1. The Synthesis of Total Amino Acids, Caffeine, and Catechins Likely Is Independently Regulated in Tender Tea Leaves

Total amino acids, caffeine, and catechins are the major non-volatile metabolites that shape tea’s taste quality, so they are the main targets for tea germplasm improvements. Previous characterizations of albino tea germplasms suggested a negative correlation between amino acids and catechins [20,21,22,23,24,25,26,27,28]. Here, we did not find such correlation by using *Baijiguan* F1 half-sibs (Figure 2a). Since the total amino acid contents are affected by not only by amino acid synthesis and degradation but also protein synthesis and degradation, it is not surprising that multiple QTLs for total amino acid contents were identified in the tea tree genome [59]. Similarly, caffeine contents were also associated with multiple QTLs [84]. The absence of tight associations among these major non-volatile components suggest that these traits are not genetically tight-linked. Thus, it is feasible to develop specialty tea cultivars with variable non-volatile contents through tea breeding. Interestingly, GA was found to be positively correlated with EGC (Figure 2a), suggesting that GA supply plays critical role in EGCG synthesis. EGCG is the dominant catechin type, and it also shows potent health benefits; there is much interest in developing specialty cultivars with higher EGCG contents. Our data suggest that GA synthesis likely represents the bottleneck for EGCG synthesis in tea. Thus, more attention should be paid to GA synthesis, especially the roles of DHQD/SD and DSDG for GA regulation [44,46]. Recently, Yang et al. (2022) identified a novel galloylglucose (1,4,6-tri-O-galloyl-β-glucopyranose, 1,4,6-TGG) from the fresh shoots of some special tea genetic resources [85], which represents a novel mechanism to regulate the free GA level in the tea tree. 1-O-galloyl-β-D-glucose (β-glucogallin) is not only the acyl donor of EGC for EGCG synthesis but also the precursor for 1,4,6-TGG synthesis [86]. The 1,4,6-TGG accumulation may result from reduced EGCG synthesis [85,87]. In addition, we found both positive and negative correlations among different catechin types (Figure 2b); this information can help us improve the total catechin contents and modify catechin compositions. 

### 3.2. Volatile Synthesis Pathways Were Metabolically Connected with Catechins or Amino Acid Metabolism

The predominant volatiles from tender fresh tea leaves are derived from the phenylpropanoid pathway (Table 1), whereas EGCG is the predominant non-volatile metabolite derived from the same pathway (Appendix A). Here, we found that the phenylpropanoid volatile pathway was positively correlated with EGCG content and that the MVA volatile pathway was positively correlated with ECG content (Figure 2a). EGCG and ECG were ranked as the top two abundant catechins in *Baijiguan* (Figure 2b). It is ideal to reduce EGCG and ECG contents for green tea germplasm breeding since higher levels of catechins are associated with a stronger astringency of tea infusion. However, this may have unintended effects on phenylpropanoid volatile and sesquiterpene volatile contents, thus negatively affecting tea’s aroma quality. The MVA volatile pathway was found to be negatively correlated with the total amino acid contents (Figure 2a). A higher free amino acid content is desirable for green tea germplasms since it endows umami taste [3,4]. However, these germplasms likely show lower MVA volatile contents and lower aroma quality. A balanced approach should be applied in tea breeding to achieve good taste quality and good aroma quality.

### 3.3. Potential Opportunities to Alter Volatile Contents and Compositions in Tender Tea Leaves

The MVA volatile synthesis pathway and fatty acid-derivative pathway are the two minor pathways contributing to the total fresh tea leaf volatile contents (Table 1). However, they also make significant contributions to tea’s aroma characteristics [83]. The positive correlations between the MVA volatile synthesis pathway and the fatty acid-derivative pathway (Figure 2a) provide opportunities to simultaneously increase their contents and breed new tea germplasms with higher aroma quality. It is important to elucidate the molecular base responsible for fatty acid degradation under normal or stress conditions, as this knowledge could lend us new tools to increase these volatile contents in fresh tea leaves.

The sesquiterpene volatile clusters were found to be highly correlated with their locations on the MVA synthesis pathway (Figure 4); similar phenomenon also was observed for the MEP pathway (Figure 5). We speculate that some unidentified biochemical or molecular mechanisms may exist and be responsible for these clustered changes. These clustered change patterns suggest that for a given volatile pathway, concurrently altering multiple volatile contents and compositions is achievable; however, it could also become a challenging issue to breakup these metabolic linkages when the breeding target is specific. 

## 4. Materials and Methods

### 4.1. Plant Material

*Camellia sinensis cv Baijiguan* is a light- and temperature-sensitive albino landrace in Fujian province of China [10,16]. F1 half-sibs were produced through natural pollination, with *Baijiguan* acting as the maternal parent. Six half-sibs showed a green leaf color, and they were numbered as follows: 0306A, 0306B, 0306H, 0306L, 0309A, and 0317D (Appendix A); the rest of the six half-sibs showed an albino leaf color similar to their maternal parent *Baijiguan*, and they were numbered as follows: 0306C, 0306D, 0306F, 0306I, 0317L, and 0317N (Appendix A). The F1 half-sibs were clonally propagated through short node cutting and grown side by side at the germplasm repository in the Tea Research Institute of Fujian Academy of Agricultural Sciences (Fu’an, Fujian, China). 

### 4.2. Tea Sample Preparation 

For volatile analysis, tender shoots including one bud and two leaves were harvested from the clonally propagated F1 population, immediately frozen in liquid nitrogen, and then stored at −80 °C in a freezer. For non-volatile analysis, after harvest, tender shoots were steamed for one minute to deactivate enzymes and then cooled down to ambient temperature. The leaf surface water was blown away by an electric fan, transferred into 100 °C oven for 1 h, and then held at 80 °C in an oven for 24 h. The dry tea was cooled down to ambient temperature, sealed in a plastic bag, and stored at 4 °C in a freezer. Before extraction, tea samples were held in an electric drying oven at 103 °C until reaching constant weight. Dry tea was ground into powder with an electric mill and then passed through a 600–1000 μm sieve to make tea powder.

### 4.3. Free Amino Acid Measurement

Free amino acids were measured following the method of GB/T 8314-2013 [88]. To create the amino acid calibration curve, theanine gradients (0, 0.2, 0.3, 0.4, 0.5, and 0.6 mg. mL^−1^) were prepared from a 10 mg. mL^−1^ stock solution. One milliliter of the theanine stock solution was transferred into a 25 mL colorimetric tube, 0.5 mL of a 1/15 mol L^−1^ phosphate buffer (PBS, pH 8.0) and a 0.5 mL of 2% (*w/v*) ninhydrin solution were added, and the tube was heated in boiling water for 15 min. After cooling down to room temperature, ddH_2_O was added to adjust the final volume to 25 mL, and the absorbance measured at 570 nm by using a PBS buffer as the blank control, the theanine calibration curve was obtained. The free amino acids from tea samples were measured with the same method, and contents calculated by using the theanine calibration curve.

### 4.4. Caffeine Quantification

Caffeine content was determined by the method of GB/T 8312-2013 [89]. One gram of tea power was weighed into a 500 mL flask, 4.5 g of magnesium oxide and 300 mL of diH_2_O were added, and then they were heated in boiling water bath for 20 min with occasional shaking. The extract was immediately filtrated into a 500 mL volumetric flask. After cooling down to room temperature, ddH_2_O was added to adjust the final volume to 500 mL. Before HPLC analysis, a 2.0 mL aliquot was filtrated through 0.45 μm membrane, 10.0 μL of the filtered sample was injected and separated with the C_18_ column (40 °C); the mobile phase was 30% (*v/v*) methanol and the speed was 1.0 mL min^−1^. A UV detector was set at 280 nm. A caffeine standard gradient (0, 10, 20, 50, and 100 μg. mL^−1^) was analyzed with abovementioned method to create calibration curve for the caffeine determination of tea samples.

### 4.5. Catechin Measurement

Catechins were quantified with the method of GB/T 8313-2018 [90]. We weighted 0.2 g of tea fine powder into a 10 mL centrifuge tube containing 5 mL of a pre-warmed (70 °C) 70% (*v/v*) methanol solution, which we immediately extracted in a 70 °C water bath for 10 min with occasional mixing. After cooling down to room temperature, the extract was centrifuged at 3500 rpm for 10 min, and the supernatant was transferred into a 10 mL volumetric flask. The pellet was re-extracted once, and the supernatant was combined. The combined supernatant volume was adjusted to 10 mL and stored at 4 °C in a freezer. Before HPLC analysis, 2.0 mL of supernatant was transferred into a 10 mL volumetric flask, the final volume was adjusted to 10 mL with a stabilization buffer (0.05% (*w/v*) EDTA, 0.05% (*w/v*) vitamin C, and 10% acetonitrile), mixed well, and then passed through 0.45 μm filter. Ten microliters of tea fusion was injected into an HPLC system and separated by a C_18_ column (250 mm × 4.6 mm × 5 μm) at 35 °C. Mobile phase A constituted 9% (*v/v*) acetonitrile, 2% (*v/v*) acetic acid, and 0.002% (*w/v*) EDTA; mobile phase B contained 80% (*v/v*) acetonitrile, 2% (*v/v*) acetic acid, and 0.002% (*w/v*) EDTA. The mobile phase speed was set to 1 mL. min^−1^. The elution program was: 0–10 min, 100% A; 10.1–25 min, isocratic to 68% A; 25.1–35 min, 68% A; 35.1–40 min, 100% A. The elute was detected by a UV detector at 278 nm. Calibration curves for GA (5–25 μg. mL^−1^), C (50–150 μg. mL^−1^), EC (50–150 μg. mL^−1^), EGC (100–300 μg. mL^−1^), ECG (50–200 μg. mL^−1^), and EGCG (100–400 μg. mL^−1^) were created via the abovementioned method and used to calculate the respective catechin content of tea samples.

### 4.6. Tea Volatile Isolation and Quantification

Tea volatile isolation was conducted following the method described before [8]. Briefly, fresh tea leaves (4.0 g) were ground into a fine powder in liquid nitrogen; then, the tea powder was transferred into a 50 mL glass tube; and 40 mL of diethyl ether, 64.5 µg of ethyl caprate, and 4.0 g of anhydrous sodium sulfate were added and extracted for 2 h under room temperature. The supernatant was distilled with an Engel apparatus to remove nonvolatile substances, and the distillate was concentrated to ~500 µL in CentriVap Console (Labconco, Kansas City, MO, USA) before GC analysis.

Tea volatile identification and quantification were performed following the method described before [8]. Each sample was analyzed with GC–MS and GC–FID (GC–MS QP2010 plus, Shimadzu, Japan). Individual volatile FID peak areas were calibrated by their RF_D_ values (relative to ethyl caprate) and then normalized to ethyl caprate peak area and sample weights.

### 4.7. Correlation Analysis

The average and standard errors were calculated with Excel software. One-way ANOVA was used to determine significance based on Duncan’s multiple range tests. Regression analysis was conducted with SPSS software (V17.0; SPSS, IBM, Armonk, NY, USA). The significance of correlations among different variables was determined via bivariate correlations based on Pearson’s correlation (two-tailed). Clustering and heat map analysis were performed with the pheatmap package in R software. For hierarchical cluster analysis, clustering algorithms chose average-linkage and distance measures using correlation distance. Pearson correlations between different metabolites or pathways were conducted with the ggcorrplot package in R software. Correlations with *p* < 0.05 were considered significant. 

## 5. Conclusions

In this study, we quantitatively measured non-volatile metabolites and volatile metabolites from 12 F1 half-sibs of *Baijiguan*, and we performed correlation analysis to explore the metabolic associations. We demonstrated that non-volatile and volatile tea metabolites are intimately connected and integrated into a larger metabolic network. Feedback inhibition or activation was implicated within individual synthesis pathways; many volatiles from the MEP or MVA pathway formed clusters and showed coordinated change patterns. We discussed these findings in the context of tea germplasm improvements. This study deepens our understanding of the metabolic connectivity among major non-volatile and volatile tea metabolites, and it provides a knowledge base for the rational design of tea tree breeding strategies for tea quality improvement.

## Figures and Tables

**Figure 1 plants-11-02557-f001:**
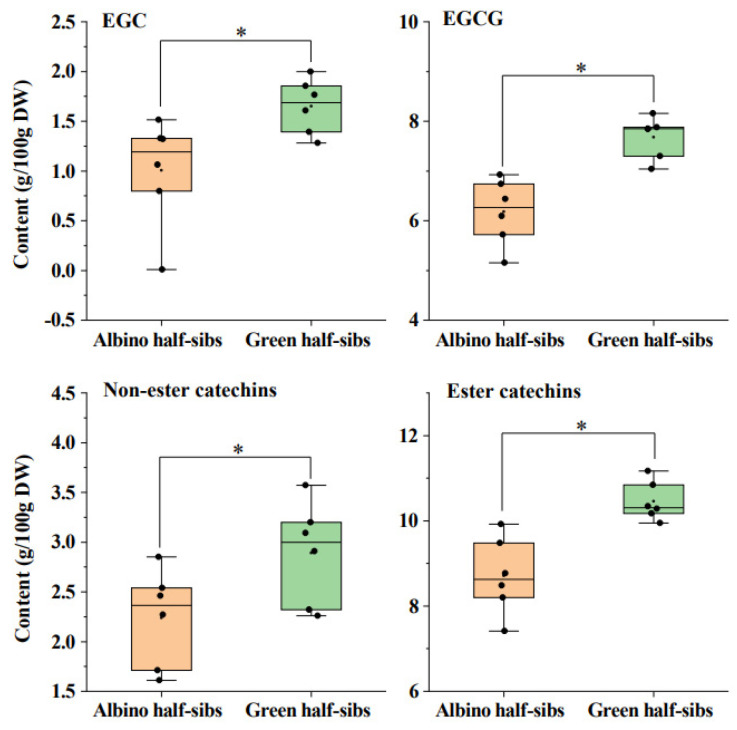
The contents of EGC, EGCG, non-ester catechins, and ester catechins from the albino half-sibs were significantly lower than those of the green half-sibs of *Camellia sinensis cv Baijiguan*. Asterisk represents statistical significance (*p* < 0.05).

**Figure 2 plants-11-02557-f002:**
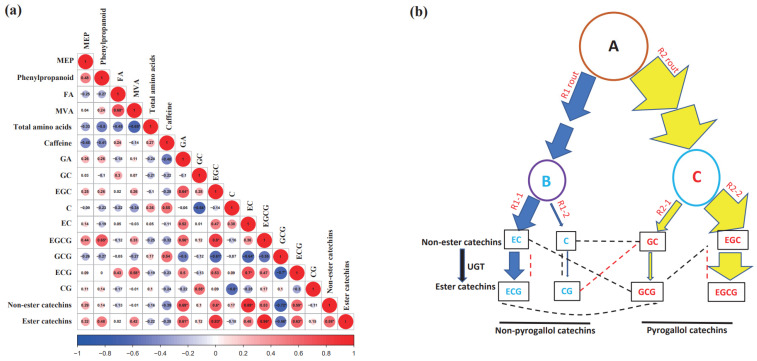
The correlations among the non-volatile metabolites and the four volatile synthesis pathways. (**a**) The correlations among total volatiles originating from four individual pathways, total amino acids, caffeine, and various catechins. (**b**) The carbon flux of catechin synthesis pathways in *Camellia sinensis cv Baijiguan* and the correlations among various catechins. A: dihydrokaempferol; B: leucocyanidin; C: leucodelphinidin. The red dashed lines represent a significant positive correlation; the black dashed lines represent a significant negative correlation. Asterisk represent statistically significant.

**Figure 3 plants-11-02557-f003:**
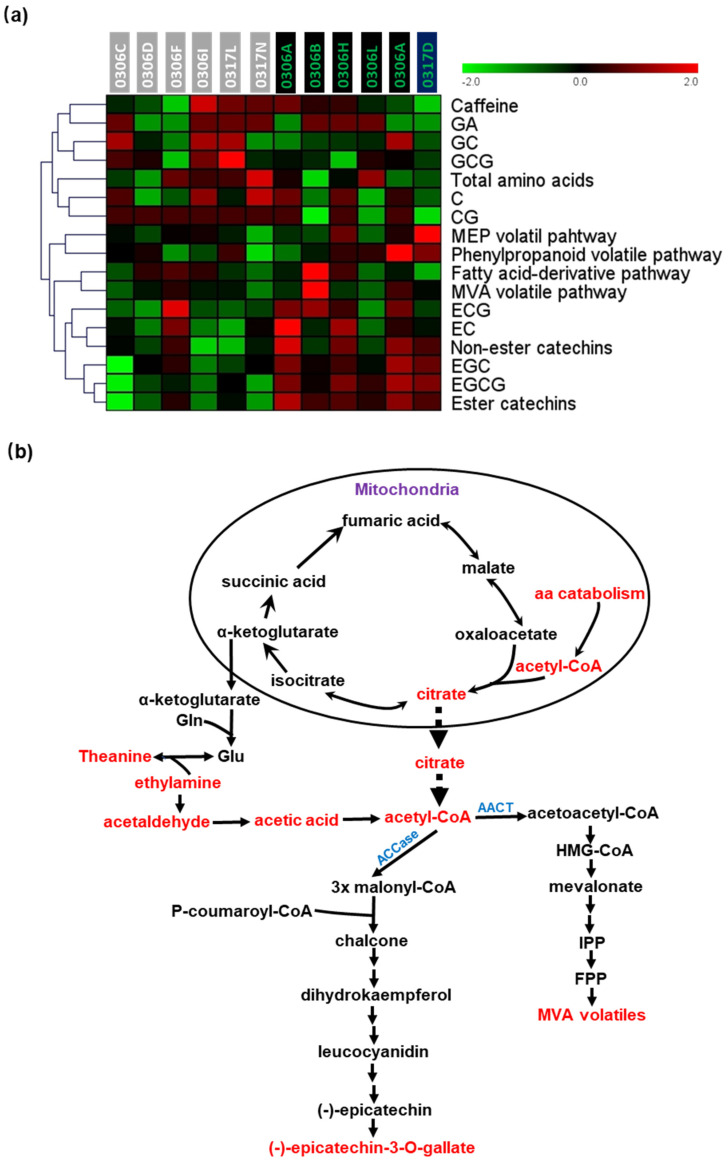
The MVA volatile synthesis pathway was positively correlated with ECG and negatively correlated with total amino acid content. (**a**) HCA and heat map analysis of volatile synthesis pathways and major non-volatile metabolites of tea; (**b**) a proposed metabolic connectivity of amino acid catabolism with the MVA volatile synthesis pathway and the catechin synthesis pathway in the tea tree.

**Figure 4 plants-11-02557-f004:**
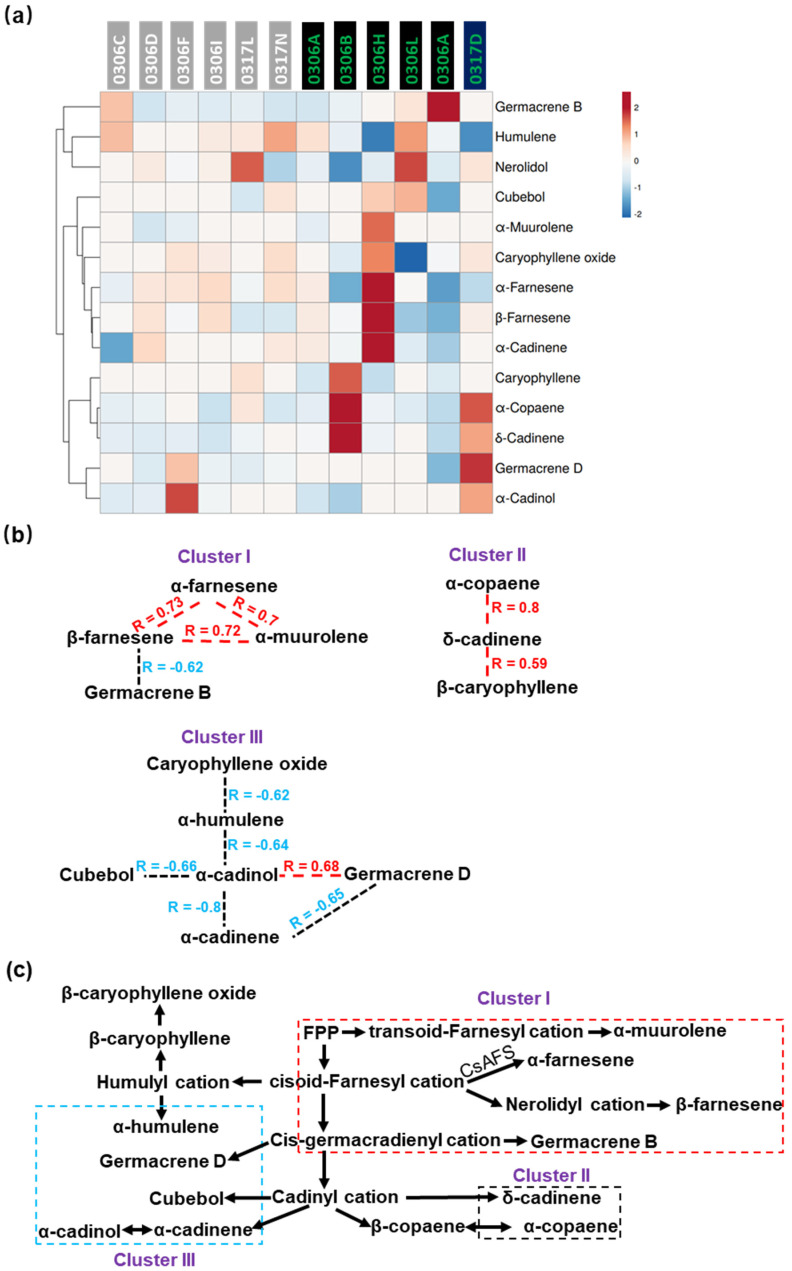
MVA volatile clustering based on correlation types and connectivity. (**a**) HCA and heat map analysis of the volatiles derived from the MVA pathway. (**b**) MVA volatiles were grouped into three clusters based on their correlations and connectivity. The red dashed lines represent a significant positive correlation; the black dashed lines represent a significant negative correlation; (**c**) proposed sesquiterpene volatile synthesis pathway.

**Figure 5 plants-11-02557-f005:**
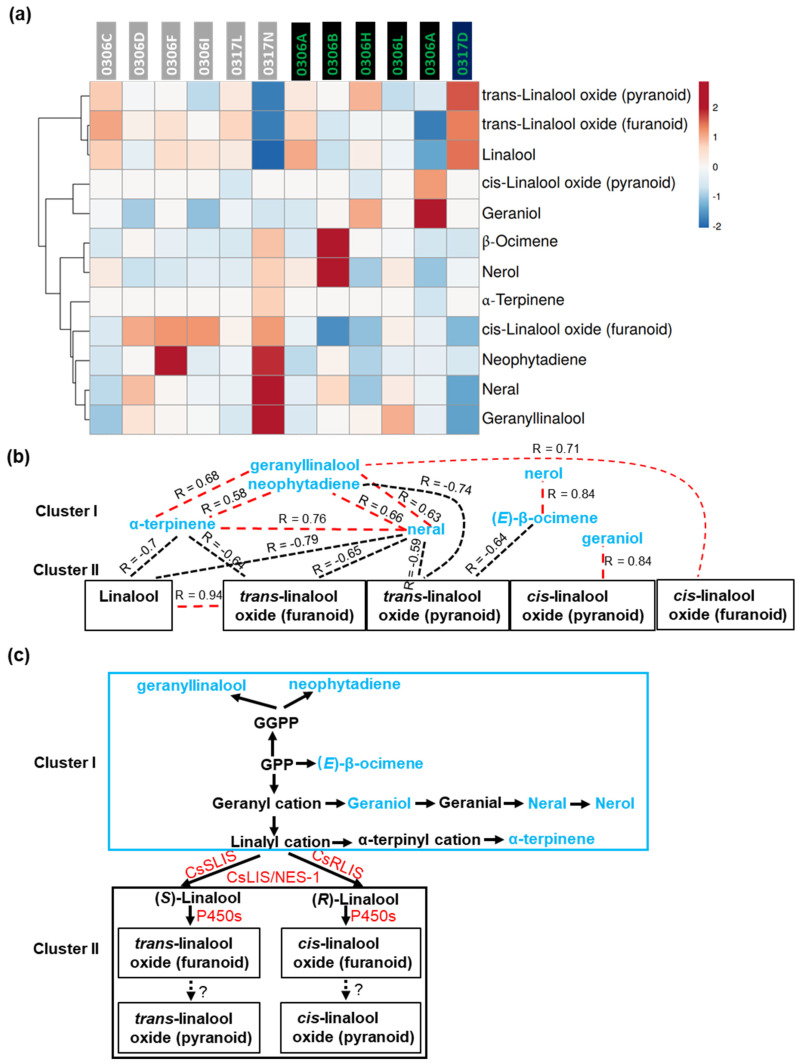
Monoterpene and diterpene volatile clustering patterns based on correlation types and connectivity. (**a**) HCA and heat map analysis of volatile metabolites of the MEP pathway. (**b**) The correlations between cluster I and cluster II metabolites of the MEP pathway. The red dashed lines represent a significant positive correlation; the black dashed lines represent a significant negative correlation. (**c**) Proposed MEP synthesis pathway.

**Figure 6 plants-11-02557-f006:**
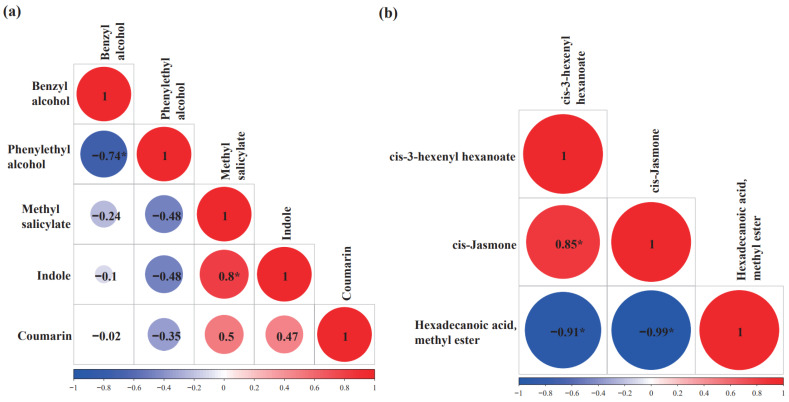
Correlations among the volatiles derived from the shikimate–phenylpropanoid pathway (**a**) and the fatty acid-derivative pathway (**b**). Asterisk represent statistically significant.

**Table 1 plants-11-02557-t001:** The fresh tea leaf volatile contents from the albino half-sibs and the green half-sibs of *Camellia sinensis cv Baijiguan* (µg. g^−^^1^ fresh weight).

Leaf Type	Germplasm Name	Phenylpropanoids	Monoterpenes/ Diterpenes	Sesquiterpenes	Fatty Acid Derivatives
Albino half-sibs	0306C	19.56 ± 1.54	12.18 ± 0.43	0.91 ± 0.12	0.19 ± 0.01
	0306D	21.18 ± 1.55	9.02 ± 0.38	1.27 ± 0.17	0.36 ± 0.02
	0306F	9.53 ± 0.88	13.21 ± 1.73	1.57 ± 0.70	0.40 ± 0.05
	0306I	14.53 ± 0.76	14.01 ± 0.45	1.26 ± 0.12	0.36 ± 0.04
	0317L	23.72 ± 1.41	11.24 ± 0.52	1.28 ± 0.09	0.23 ± 0.01
	0317N	4.35 ± 0.11	3.06 ± 0.12	0.75 ± 0.03	0.15 ± 0.01
Green half-sibs	0306A	11.48 ± 0.61	10.24 ± 0.57	1.14 ± 0.06	0.26 ± 0.01
	0306B	17.58 ± 0.58	8.79 ± 0.78	3.19 ± 0.32	0.72 ± 0.05
	0306H	23.06 ± 0.59	18.53 ± 0.67	1.11 ± 0.09	0.39 ± 0.02
	0306L	22.72 ± 1.61	7.24 ± 0.60	0.89 ± 0.04	0.15 ± 0.00
	0309A	37.24 ± 0.56	14.57 ± 0.39	1.71 ± 0.15	0.27 ± 0.02
	0317D	28.26 ± 1.35	31.00 ± 1.82	1.33 ± 0.33	0.06 ± 0.00

Data are expressed as mean ± standard error (*n* = 4).

## Data Availability

Data generated with this study are available in the main text, Appendix A, and the Appendix A.

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
