# Peer review of "Metabolomic Profiling in Combination with Data Association Analysis Provide Insights about Potential Metabolic Regulation Networks among Non-Volatile and Volatile Metabolites in Camellia sinensis cv Baijiguan"

_plants, 2022, doi:10.3390/plants11192557_

Round 1

Reviewer 1 Report

Chen’ s article used metabolite analysis and metabolic biosynthetic pathway to investigate the correlation between different biosynthetic routes (or cluster mentioned in the manuscript). Aims to elucidate the relation between the phenotype (albino tea and green tea) and chemotype, and it will b helpful for tea breeding. However, the article requires aggressive revision due to multiple grammar error and the ambiguous sentences. I recommend use editing service or proofreading by native speaker to address this issue. Some suggestions for minor editing are mentioned below:

(1)   Please unify the font used in reference list

(2)   I suggest marking the phenotype of each sample on figure 3A, 4A, 5A.

(3)   The font size of figure 6 is too small to read

(4)   Please make sure all the sentences are not misleading:
in line 285-287, the authors concluded that the linalool synthases efficiently transfer carbon into linalool accumulation, but there are some questions. (A)is CsRIS and CsRLIS are the only way to produce linalool in this spcies? (B) it is correct that the reference cited is associated to the CsSRIS and CsRLIS functional characterization, but it Is not equivalent to the sentence “From the stable levels of individual volatile contents presented in Supplementary data 2, we found that about 60-90% of carbon was channeled into linalool synthesis [20]”

(5)   The clustering method is not described properly in the manuscript. In figure5A, the patterns could be grouped in to three subgroups, and it properly due to they are biosynthesize via three different routes (monoterpene, sesquiterpene and diterpene)

(6)   In final, I wonder if this article can provide a overall TCA analysis that can provide summary of differences between albino and green half-sib, which is difficult to comprehend from current manuscript.

Author Response

Reviewer 1

Comments and Suggestions for Authors

Chen’ s article used metabolite analysis and metabolic biosynthetic pathway to investigate the correlation between different biosynthetic routes (or cluster mentioned in the manuscript). Aims to elucidate the relation between the phenotype (albino tea and green tea) and chemotype, and it will b helpful for tea breeding. However, the article requires aggressive revision due to multiple grammar error and the ambiguous sentences. I recommend use editing service or proofreading by native speaker to address this issue. Some suggestions for minor editing are mentioned below:

Authors: I asked a native speaker for a proofreading, and the revisions were highlighted, thank you!

  • Please unify the font used in reference list

Authors: The reference list was revised according to journal style, thanks!

  • I suggest marking the phenotype of each sample on figure 3A, 4A, 5A.

Authors: The sample labeling from Figure 3A, 4A, and 5A were revised such that the font color was in accordance with respective leaf color: white font color represent albino leaf germplasms and green font color represent green leaf germplasms.

  • The font size of figure 6 is too small to read

Authors: The font size of Figure 6 was enlarged, thanks!

  • Please make sure all the sentences are not misleading:
    in line 285-287, the authors concluded that the linalool synthases efficiently transfer carbon into linalool accumulation, but there are some questions. (A)is CsRIS and CsRLIS are the only way to produce linalool in this spcies? (B) it is correct that the reference cited is associated to the CsSRIS and CsRLIS functional characterization, but it Is not equivalent to the sentence “From the stable levels of individual volatile contents presented in Supplementary data 2, we found that about 60-90% of carbon was channeled into linalool synthesis [20]”

Authors: Liu et al (2017) reported that a splicing isoform CsLIS/NES-1 also was capable to catalyze linalool synthesis (DOI: 10.1111/pce.13080). Thus, we revised this statement in the text and added CsLIS/NES-1 into Figure 5c. The reference 20 was removed from the text to avoid potential misunderstanding, thank you!

  • The clustering method is not described properly in the manuscript. In figure5A, the patterns could be grouped in to three subgroups, and it properly due to they are biosynthesize via three different routes (monoterpene, sesquiterpene and diterpene)

Authors: we revised the clustering method description (lines 460-463). In Figure 5a, cis-linalool oxide (pyranoid) and geraniol were grouped into one subgroup, this subgroup showed closer distance with other upstream volatiles (Figure 5a). Interestingly, cis-linalool oxide and trans-linalool oxide showed quite different correlations with other intermediates of MEP pathway (Figure 5b), this may reflect their unique role in MEP pathway regulation. We agree with the reviewer that the origin (synthesis routs) seem have some impacts on their grouping patterns as we showed in Figure 5a-5c.

  • In final, I wonder if this article can provide a overall TCA analysis that can provide summary of differences between albino and green half-sib, which is difficult to comprehend from current manuscript.

Authors: That’s a great question! The non-volatile metabolites measured in this study are mainly related with tea quality traits including total amino acids, caffeine, and catechins. As such, the TCA cycle intermediates as well as individual amino acid profiles were not resolved, it remains an open question whether TCA cycle flux be affected by leaf color mutation. TCA cycle not only provides intermediates for amino acid synthesis, it also is involved in aspartate derived amino acid degradation, including lysine, methionine, threonine and isoleucine. In contrast to our assumption, there was no statistically significant difference of the total amino acid contents between the albino half sibs and the green half sibs. A systemic amino acid and organic acid profiling in the future could provide clues regarding the TCA operation modes in albino leaves and green leaves. Thank you!

Reviewer 2

Comments and Suggestions for Authors

Dear Authors,

The research paper entitled "Metabolomics Profiling in Combination with Data Association Analysis 2 Provide Insights about Potential Metabolic Regulation Networks among 3 Non-volatile and Volatile Metabolites in Camellia sinensis cv Baijiguan", has a valuable approach, with high originality and complexity, adding important scientific value. However, I do recommend a minor revision prior publication.

Authors: Thank you very much for your comments!

Comments:

Highlights should be improved being more specific on the results obtained, e.g. percentages, comparison between types of samples, etc.

Authors: we revised the abstract to include the main results obtained, thank you!

The introduction section – please describe and discuss the unique features of this research in the context of future applications and their influence.

Authors: We describe and discuss the unique features of this research in the introduction section (lines 83-84), then briefly mention its future applications in lines 87-89. In following discussion section (lines 350-389) we further discuss the potential applications and influence of this research for future tea germplasm improvements. We also add additional wordings in the discussion section (lines 360-362; line 382-383; lines 386-389).

 The methodology section – well described, however in sections 4.3, 4.4 and 4.5 please add the reference of the method used

Authors: The reference for section 4.3, 4.4, and 4.5 were added into the reference list, thanks!

All the results are systematically presented and discussed, but their implications should be detailed in the broadest context possible. There is a lack of limitations of the work described (for all types of results systematically).

Authors: We add additional contents in the discussion section (lines 360-362; line 382-383; lines 386-389) to discuss the implications of this study, thank you!

In the conclusion section, more practical applications should be depicted from the results obtained.

Authors: We added more practical applications into the conclusion section in the context of tea breeding for metabolic trait improvements, thanks!

Reviewer 2 Report

Dear Authors,

The research paper entitled "Metabolomics Profiling in Combination with Data Association Analysis 2 Provide Insights about Potential Metabolic Regulation Networks among 3 Non-volatile and Volatile Metabolites in Camellia sinensis cv Baijiguan", has a valuable approach, with high originality and complexity, adding important scientific value. However, I do recommend a minor revision prior publication.

Comments:

Highlights should be improved being more specific on the results obtained, e.g. percentages, comparison between types of samples, etc.

The introduction section – please describe and discuss the unique features of this research in the context of future applications and their influence.

The methodology section – well described, however in sections 4.3, 4.4 and 4.5 please add the reference of the method used

All the results are systematically presented and discussed, but their implications should be detailed in the broadest context possible. There is a lack of limitations of the work described (for all types of results systematically). 

In the conclusion section, more practical applications should be depicted from the results obtained.

Author Response

Comments and Suggestions for Authors

Dear Authors,

The research paper entitled "Metabolomics Profiling in Combination with Data Association Analysis 2 Provide Insights about Potential Metabolic Regulation Networks among 3 Non-volatile and Volatile Metabolites in Camellia sinensis cv Baijiguan", has a valuable approach, with high originality and complexity, adding important scientific value. However, I do recommend a minor revision prior publication.

Authors: Thank you very much for your comments!

Comments:

Highlights should be improved being more specific on the results obtained, e.g. percentages, comparison between types of samples, etc.

Authors: we revised the abstract to include the main results obtained, thank you!

The introduction section – please describe and discuss the unique features of this research in the context of future applications and their influence.

Authors: We describe and discuss the unique features of this research in the introduction section (lines 83-84), then briefly mention its future applications in lines 87-89. In following discussion section (lines 350-389) we further discuss the potential applications and influence of this research for future tea germplasm improvements. We also add additional wordings in the discussion section (lines 360-362; line 382-383; lines 386-389).

 The methodology section – well described, however in sections 4.3, 4.4 and 4.5 please add the reference of the method used

Authors: The reference for section 4.3, 4.4, and 4.5 were added into the reference list, thanks!

All the results are systematically presented and discussed, but their implications should be detailed in the broadest context possible. There is a lack of limitations of the work described (for all types of results systematically).

Authors: We add additional contents in the discussion section (lines 360-362; line 382-383; lines 386-389) to discuss the implications of this study, thank you!

In the conclusion section, more practical applications should be depicted from the results obtained.

Authors: We added more practical applications into the conclusion section in the context of tea breeding for metabolic trait improvements, thanks!